# META LEARNING VIA LEARNED LOSS

## ABSTRACT

We present a meta-learning method for learning parametric loss functions that can generalize across different tasks and model architectures. We develop a pipeline for training such loss functions, targeted at maximizing the performance of model learning with them. We observe that the loss landscape produced by our learned losses significantly improves upon the original task-specific losses in both supervised and reinforcement learning tasks. Furthermore, we show that our meta-learning framework is flexible enough to incorporate additional information at *meta-train* time. This information shapes the learned loss function such that the environment does not need to provide this information during *meta-test* time.

## 1 INTRODUCTION

Inspired by the remarkable capability of humans to quickly learn and adapt to new tasks, the concept of learning to learn, or *meta-learning*, recently became popular within the machine learning community (Andrychowicz et al., 2016; Duan et al., 2016; Finn et al., 2017). We can classify learning-to-learn methods into roughly 2 categories: approaches that lead to learning representations that can generalize and are easily adaptable to new tasks (Finn et al., 2017), and learning approaches that attempt to learn how to optimize models (Andrychowicz et al., 2016; Duan et al., 2016). In this paper we investigate the second type of approach and propose a learning framework that is able to learn loss function representations that can then be used to optimize models for new tasks.

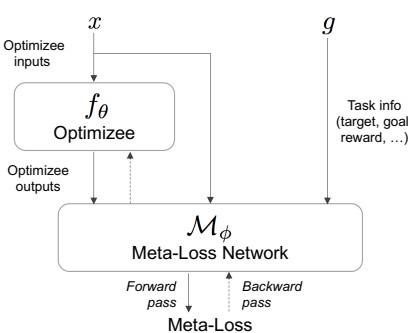

Figure 1: Framework overview

Specifically, the purpose of this work is to encode learning strategies into an adaptive high-dimensional loss function, or a *meta-loss*, which generalizes across multiple training contexts or tasks. Inspired by *inverse reinforcement learning* (Ng et al., 2000), our work combines the *learning to learn* paradigm of meta-learning with the generality of learning loss landscapes. We construct a unified, fully differentiable framework that can learn model-agnostic loss functions to provide a strong learning signal for a large range of model classes, such as classifiers, regressors or control policies.

The contributions of this work are as follows: i) we present a framework for learning adaptive, high-dimensional loss functions through back-propagation that shape the loss landscape such that it can be efficiently optimized with gradient descent. This framework involves an inner and an outer loop. In the inner loop, a model or an *optimizee* is trained with gradient descent using the loss coming from our learned meta-loss function. Fig. 1 shows the pipeline for updating the optimizee with the meta-loss. The outer loop optimizes the meta-loss function by minimizing a *task-loss*, such as a standard regression or reinforcement-learning loss, that is induced by the updated optimizee. We show that our learned meta-loss functions improves over directly learning via the task-loss itself while maintaining the generality of the task-loss. ii) We show how we can utilize extra information that helps shape the loss landscapes at *meta-train* time. This extra information can take on various forms, such as exploratory signals or expert demonstrations for RL tasks. After training the meta-loss function, the task-specific losses are no longer required since the training of optimizees can be performed entirely by using the meta-loss function alone, without requiring the extra information given at *meta-train* time. In this way, our meta-loss can find more efficient ways to optimize the original task loss.

## 2 RELATED WORK

Meta-learning originates in the concept of learning to learn (Schmidhuber, 1987; Bengio & Bengio, 1990; Thrun & Pratt, 2012). Let us consider gradient based learning approaches, that update the parameters of an *optimizee* $f_\theta(x)$, with model parameters $\theta$ and inputs $x$ as follows:

$$\theta_{\text{new}} = h_\psi(\theta, \nabla_\theta \mathcal{L}_\phi(y, f_\theta(x))); \tag{1}$$

where we take the gradient of a loss function $\mathcal{L}$, parametrized by $\phi$, with respect to the optimizee's parameters $\theta$ and use a gradient transform $h$, parametrized by $\psi$, to compute new model parameters $\theta_{\text{new}}$[1]. In this context, we can divide related work on meta-learning into learning model parameters $\theta$ that can be easily adapted to new tasks (Finn et al., 2017; Mendonca et al., 2019; Gupta et al., 2018; Yu et al., 2018), learning optimizer policies $h$ that transform parameters updates with respect to known loss or reward functions (Maclaurin et al., 2015; Andrychowicz et al., 2016; Li & Malik, 2016; Franceschi et al., 2017; Meier et al., 2018; Duan et al., 2016), or learning loss/reward function representations $\phi$ (Sung et al., 2017; Houthooft et al., 2018; Zou et al., 2019). Alternatively, in unsupervised learning settings, meta-learning has been used to learn unsupervised rules that can be transferred between tasks (Metz et al., 2019; Hsu et al., 2018)

Our framework falls into the category of learning loss landscapes. Similar to works by Sung et al. (2017) and Houthooft et al. (2018), we aim at learning loss function parameters $\phi$ that can be applied to various optimizee models, e.g. regressors, classifiers or agent policies. Our learned loss functions are independent of the model parameters $\theta$ that are to be optimized, thus they can be easily transferred to other optimizee models. This is in contrast to methods that meta-learn model-parameters $\theta$ directly, such as (Finn et al., 2017; Mendonca et al., 2019), where the learned representation $\theta$ can not be separated from the original model of the optimizee. The idea of learning loss landscapes or reward functions in the reinforcement learning (RL) setting can be traced back to the field of inverse reinforcement learning (Ng et al., 2000; Abbeel & Ng, 2004, IRL). However, in contrast to IRL we do not require expert demonstrations (however we can incorporate them). Instead we use task losses as a measure of the effectiveness of our loss function when using it to optimize an optimizee, and as the optimization objective for $\phi$ (parameters of the learned loss).

Closest to our method are the works on *evolved policy gradients* (Houthooft et al., 2018), *teacher networks* (Wu et al., 2018), *meta-critics* (Sung et al., 2017) and *meta-gradient RL* (Xu et al., 2018). In contrast to using an evolutionary approach (e.g. Houthooft et al., 2018), we design a differentiable framework and describe a way to optimize the loss function with gradient descent in both supervised and reinforcement learning settings. Wu et al. (2018) propose that instead of learning a differentiable loss function directly, a teacher network is trained to predict parameters of a manually designed loss function, which requires a new teacher network design and training for each new loss function class. In Xu et al. (2018), discount and bootstrapping parameters are learned online to optimize a task-specific meta-objective. Our method does not require manual design of the loss function parameterization or choosing particular parameters that have to be optimized, as our loss functions are learned entirely from data. Finally, in work by Sung et al. (2017) a *meta-critic* is learned to provide a task-conditional value function, used to train an actor policy. Although training a meta-critic in the supervised setting reduces to learning a loss function as in our work, in the reinforcement learning setting we show that it is possible to use learned loss functions to optimize policies directly with gradient descent.

## 3 META-LEARNING VIA LEARNED LOSS

In this work, we aim to learn a loss function, which we call *meta-loss*, that is used to train an *optimizee*, e.g. a classifier, a regressor or an agent policy. More concretely, we aim to learn a meta-loss function $\mathcal{M}_\phi$ with parameters $\phi$, which can be used to predict the loss value $\mathcal{L}_{\text{learned}}$ that can be used to train an optimizee $f_\theta$ with parameters $\theta$ via gradient descent:

$$\theta_{\text{new}} = \theta - \alpha \nabla_\theta \mathcal{L}_{\text{learned}}, \text{ where } \mathcal{L}_{\text{learned}} = \mathcal{M}_\phi(y, f_\theta(x)) \tag{2}$$

where $y$ can be ground target information in supervised learning settings or goal and state information for reinforcement learning settings. In short, we aim to learn a loss function that can be used as

---

[1]for simple gradient descent $h(\theta, \nabla_\theta \mathcal{L}(y, f_\theta(x))) = \theta - \psi \nabla_\theta \mathcal{L}(y, f_\theta(x))$

| **Algorithm 1** ML$^3$ at (*meta-train*) | **Algorithm 2** ML$^3$ at (*meta-test*) |
|---|---|
| 1: $\phi, \theta \leftarrow$ randomly initialize | 1: $M \leftarrow$ # of optimizee updates |
| 2: **while** not done **do** | 2: $\theta \leftarrow$ randomly initialize |
| 3: $\quad x, y \leftarrow$ Sample task samples from $\mathcal{T}$ | 3: **for** $j \in \{0, \dots, M\}$ **do** |
| 4: $\quad \mathcal{L}_{\text{learned}} = \mathcal{M}(y, f_\theta(x))$ | 4: $\quad x, y \leftarrow$ Sample task samples from $\mathcal{T}$ |
| 5: $\quad \theta_{\text{new}} \leftarrow \theta - \alpha \nabla_\theta \mathbb{E}_x [\mathcal{L}_{\text{learned}}]$ | 5: $\quad \mathcal{L}_{\text{learned}} = \mathcal{M}(y, f_\theta(x))$ |
| 6: $\quad \phi \leftarrow \phi - \eta \nabla_\phi \mathcal{L}_\mathcal{T}(y, f_{\theta_{\text{new}}})$ | 6: $\quad \theta \leftarrow \theta - \alpha \nabla_\theta \mathbb{E}_x [\mathcal{L}_{\text{learned}}]$ |

depicted in Algorithm 2. Towards this goal, we propose an algorithm to train learning loss function parameters $\phi$ via gradient descent. During *meta-train* time the goal is to train the loss function parameters $\phi$ such that it successfully optimizes a model $f_\theta$ for chosen task(s) $\mathcal{T}$. The key challenge is to derive a training signal for loss parameters $\phi$. In the following, we describe our approach to addressing this challenge, which we call **M**eta-**L**earning via **L**earned **L**oss (ML$^3$).

### 3.1 ML$^3$ FOR SUPERVISED LEARNING

We start with supervised learning settings, in which our framework aims at learning a meta-loss function $\mathcal{M}_\phi(y, f_\theta(x))$ that predicts the loss value given the ground truth target $y$ and the predicted target $f_\theta(x)$. For clarity purposes we constrain the following presentation to learning a meta-loss network that predicts the loss value for training a regressor $f_\theta$ via gradient descent, however the methodology trivially generalizes to classification tasks.

Our meta-learning framework starts with randomly initialized model parameters $\theta$ and loss parameters $\phi$. The current loss parameters are then used to predict loss value $\mathcal{L}_{\text{learned}} = \mathcal{M}_\phi(y, f_\theta(x))$. To optimize model parameters $\theta$ we need to compute the gradient of the loss value with respect to $\theta$, $\nabla_\theta \mathcal{L} = \nabla_\theta \mathcal{M}_\phi(y, f_\theta(x))$. Using the chain rule, we can decompose the gradient computation into the gradient of the loss network with respect to predictions of model $f_\theta(x)$ times the gradient of model $f$ with respect to model parameters[2],

$$\nabla_\theta \mathcal{M}_\phi(y, f_\theta(x)) = \nabla_f \mathcal{M}_\phi(y, f_\theta(x)) \nabla_\theta f_\theta(x). \tag{3}$$

Once we have updated the model parameters $\theta_{\text{new}} = \theta - \alpha \nabla_\theta \mathcal{L}_{\text{learned}}$ using the current meta-loss network parameters $\phi$, we want to measure how much learning progress has been made with loss-parameters $\phi$ and optimize $\phi$ via gradient descent. Note, that the new model parameters $\theta_{\text{new}}$ are implicitly a function of loss-parameters $\phi$, because changing $\phi$ would lead to different $\theta_{\text{new}}$. In order to evaluate $\theta_{\text{new}}$, and through that loss-parameters $\phi$, we introduce the notion of a *task-loss* during *meta-train* time. For instance, we use the Mean-Squared-Error loss, which is typically used for regression tasks, as a task-loss $\mathcal{L}_\mathcal{T} = (y - f_{\theta_{\text{new}}}(x))^2$. We now optimize loss parameters $\phi$ by taking the gradient of $\mathcal{L}_\mathcal{T}$ with respect to $\phi$ as follows[2]:

$$\nabla_\phi \mathcal{L}_\mathcal{T}(y, f_{\theta_{\text{new}}}(x)) = \nabla_{\theta_{\text{new}}} \mathcal{L}_\mathcal{T} \nabla_\phi \theta_{\text{new}} \tag{4}$$

$$= \nabla_{\theta_{\text{new}}} \mathcal{L}_\mathcal{T} \nabla_\phi [\theta - \alpha \nabla_\theta \mathbb{E}[\mathcal{M}_\phi(x, f_\theta(x))] \tag{5}$$

Optimization of the loss-parameters can either happen after *each inner gradient* step (where inner refers to using the current loss parameters to update $\theta$), or after $M$ *inner gradient steps* with the current meta-loss network $\mathcal{M}_\phi$. The latter option requires back-propagation through a chain of all optimizee update steps. In practice we notice that updating the meta-parameters $\phi$ after each inner gradient update step works better. We summarize the *meta-train* phase in Algorithm 1, with one *inner* gradient step.

### 3.2 ML$^3$ REINFORCEMENT LEARNING

In this section, we introduce several modifications that allow us to apply the ML$^3$ framework to reinforcement learning problems. Let $\mathcal{M} = (S, A, P, R, p_0, \gamma, T)$ be a finite-horizon Markov Decision Process (MDP), where $S$ and $A$ are state and action spaces, $P : S \times A \times S \to \mathbb{R}_+$ is a state-transition probability function or system dynamics, $R : S \times A \to \mathbb{R}$ a reward function, $p_0 : S \to \mathbb{R}_+$ an initial state distribution, $\gamma$ a reward discount factor, and $T$ a horizon. Let

---

[2]alternatively this gradient computation can performed using automatic differentiation

$\tau = (s_0, a_0, \ldots, s_T, a_T)$ be a trajectory of states and actions and $R(\tau) = \sum_{t=0}^{T-1} \gamma^t R(s_t, a_t)$ the trajectory return. The goal of reinforcement learning is to find parameters $\theta$ of a policy $\pi_\theta(a|s)$ that maximizes the expected discounted reward over trajectories induced by the policy: $\mathbb{E}_{\pi_\theta}[R(\tau)]$ where $s_0 \sim p_0, s_{t+1} \sim P(s_{t+1}|s_t, a_t)$ and $a_t \sim \pi_\theta(a_t|s_t)$. In what follows, we show how to train a meta-loss network to perform effective policy updates in a reinforcement learning scenario. To apply our ML$^3$ framework, we replace the optimizee $f_\theta$ from the previous section with a stochastic policy $\pi_\theta(a|s)$. We present two applications of ML$^3$ to RL.

### 3.2.1 ML$^3$ FOR MODEL-BASED REINFORCEMENT LEARNING

Model-based RL (MBRL) attempts to learn a policy $\pi$ by first learning a dynamic model $P$. Intuitively, if the model $P$ is accurate, we can use it to optimize the policy parameters $\theta$. As we typically do not know the dynamics model a-priori, MBRL algorithms iterate between using the current approximate dynamics model $P$, to optimize the policy $\pi$ such that it maximizes the reward $R$ under $P$, then use the optimized policy $\pi$ to collect more data which is used to update the model $P$. In this context, we aim to learn a loss function that is used to optimize policy parameters through our meta-network $\mathcal{M}$.

Similar to the supervised learning setting we use current meta-parameters $\phi$ to optimize policy parameters $\theta$ under the current dynamics model $P$: $\theta_{\text{new}} = \theta - \alpha \nabla_\theta [\mathcal{M}_\phi(\tau, g)]$, where $\tau = (s_0, a_0, \ldots, s_T, a_T)$ is the sampled trajectory and the variable $g$ captures some task-specific information, such as the goal state of the agent.

To optimize $\phi$ we again need to define a task loss, which in the MBRL setting can be defined as $\mathcal{L}_{\mathcal{T}}(g, \pi_{\theta_{\text{new}}}) = -\mathbb{E}_{\pi_{\theta_{\text{new}}}, P}[R_g(\tau_{\text{new}})]$, denoting the reward that is achieved under the current dynamics model $P$. To update $\phi$, we compute the gradient of the task loss $\mathcal{L}_{\mathcal{T}}$ wrt. $\phi$, which involves differentiating all the way through the reward function, dynamics model and the policy that was updated using the meta-loss $\mathcal{M}_\phi$. The pseudo-code in Algorithm 3 (Appendix A.1) illustrates the MBRL learning loop. In Algorithm 5 (Appendix A.1), we show the policy optimization procedure during meta-test time. Notably, we have found that in practice, the model of the dynamics $P$ is not needed anymore for policy optimization at meta-test time. The meta-network learns to implicitly represent the gradients of the dynamics model and can produce a loss to optimize the policy directly.

### 3.2.2 ML$^3$ FOR MODEL-FREE REINFORCEMENT LEARNING

Finally, we consider the model-free reinforcement learning (MFRL) case, where we learn a policy without learning a dynamics model. In this case, we can define a surrogate objective, which is independent of the dynamics model, as our task-specific loss (Williams, 1992; Sutton et al., 2000; Schulman et al., 2015):

$$\mathcal{L}_{\mathcal{T}}(g, \pi_{\theta_{\text{new}}}) = -\mathbb{E}_{\pi_{\theta_{\text{new}}}}[R_g(\tau_{\text{new}}) \log \pi_{\theta_{\text{new}}}(\tau_{\text{new}})] = -\mathbb{E}_{\pi_{\theta_{\text{new}}}}\left[R_g(\tau_{\text{new}}) \sum_{t=0}^{T-1} \log \pi_{\theta_{\text{new}}}(a_t|s_t)\right] \quad (6)$$

Similar to the MBRL case, the task loss is indirectly a function of the meta-parameters $\phi$ that are used to update the policy parameters. Although we are evaluating the task loss on full trajectory rewards, we perform policy updates from Eq. 2 using stochastic gradient descent (SGD) on the meta-loss with mini-batches of experience $(s_i, a_i, r_i)$ for $i \in \{0, \ldots, B-1\}$ with batch size $B$, similar to Houthooft et al. (2018). The inputs of the meta-loss network are the sampled states, sampled actions, task information $g$ and policy probabilities of the sampled actions: $\mathcal{M}_\phi(s, a, \pi_\theta(a|s), g)$. In this way, we enable efficient optimization of very high-dimensional policies with SGD provided only with trajectory-based rewards. In contrast to the above MBRL setting, the rollouts used for task-loss evaluation are real system rollouts, instead of simulated rollouts. At test time, we use the same policy update procedure as in the MBRL setting, see Algorithm 5 (Appendix A.1).

### 3.3 THE TASK LOSS AND ADDING EXTRA INFORMATION DURING *meta-train*

So far, we have discussed using standard task losses, such as MSE-loss for regression or reward functions for RL settings. However, it is possible to provide more information about the task at *meta-train* time, which can influence the learning of the loss-landscape. In our work, we experiment with 3 different types of extra information at *meta-train* time: for supervised learning we show that providing ground truth information about the optimal parameters can help shape a convex loss-landscape for otherwise non-convex optimization problems; for reinforcement learning tasks we demonstrate that

by providing additional rewards in the task loss during meta-train time, we can encourage the trained meta-loss to learn exploratory behaviors; and finally also for reinforcement learning tasks we also show how expert demonstrations can be incorporated to learn loss functions which can generalize to new tasks. In all these settings, the additional information shapes the learned loss function such that the environment does not need to provide this information during meta-test time.

## 4 EXPERIMENTS

In this section we evaluate the applicability and the benefits of the learned meta-loss from two different view points. First, we study the benefits of using standard task losses, such as the mean-squared error loss for regression, to train the meta-loss in Section 4.1. We analyze how a learned meta-loss compares to using a standard task-loss in terms of generalization properties and convergence speed. Second, we study the benefit of adding extra information at *meta-train* time in Section 4.2.

### 4.1 LEARNING TO MIMIC AND IMPROVE OVER KNOWN TASK LOSSES

First, we analyze how well our meta-learning framework can learn to mimic and improve over standard task losses for both supervised and reinforcement learning settings. For these experiments, the meta-network is parameterized by a neural network with two hidden layers of 40 neurons each.

#### 4.1.1 META-LOSS FOR SUPERVISED LEARNING

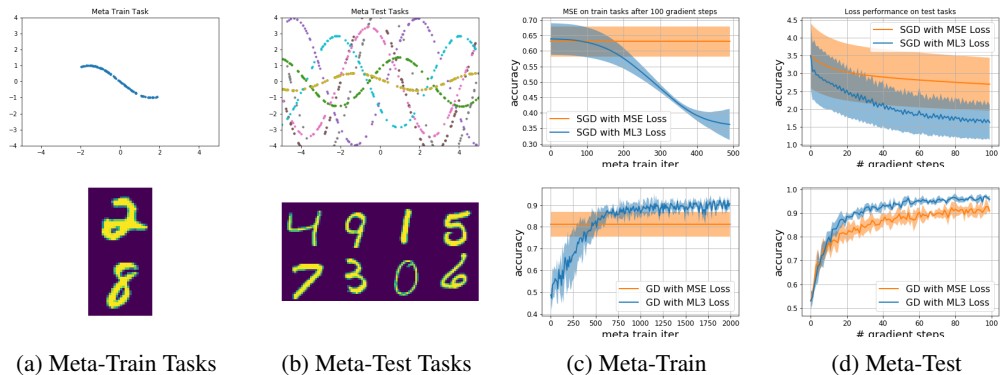

(a) Meta-Train Tasks     (b) Meta-Test Tasks     (c) Meta-Train     (d) Meta-Test

Figure 2: Meta-learning for regression (top) and binary classification (bottom) tasks. (a) meta-train task, (b) meta-test tasks, (c) performance of the meta-network on the meta-train task as a function of (outer) meta-train iterations in blue, as compared to SGD using the task-loss directly in orange, (d) average performance of meta-loss on meta-test tasks as a function of the number of gradient update steps

In this set of experiments, we evaluate how well our meta-learning framework can learn loss functions $\mathcal{M}_\phi$ for regression and classification tasks. In particular, we perform experiments on sine function regression and binary classification of digits (see details in Appendix A.4). At meta-train time, we randomly draw one task for meta-training (see Fig. 2 (a)), and at meta-test time we randomly draw 10 test tasks for regression, and 4 test tasks for classification (Fig. 2(b)). We compare the performance of using SGD with the task-loss $\mathcal{L}$ directly (in orange) to SGD using the learned meta-network $\mathcal{M}$ (in blue), both using a learning rate $\alpha = 0.001$. In Fig. 2 (c) we show the average performance of the meta-network $\mathcal{M}_\phi$ as it is being learned, as a function of (outer) meta-train iterations in blue. In both regression and classification tasks, the meta-loss eventually leads to a better performance on the meta-train task as compared to the task loss. In Fig. 2 (d) we evaluate SGD using $\mathcal{M}_\phi$ vs SGD using $\mathcal{L}$ on previously unseen (and out-of-distribution) meta-test tasks as a function of the number of gradient steps. Even on these novel test tasks, our learned $\mathcal{M}_\phi$ leads to improved performance as compared to the task-loss.

#### 4.1.2 LEARNING REWARD FUNCTIONS FOR MODEL-BASED REINFORCEMENT LEARNING

In the MBRL example, the tasks consist of a free movement task of a point mass in a 2D space, we call this environment PointmassGoal, and a reaching task with a 2-link 2D manipulator, which we

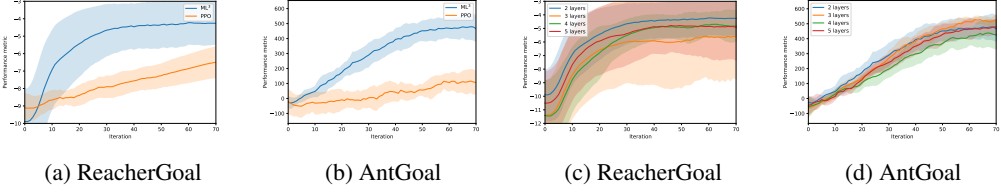

(a) Generalization Pointmass    (b) Meta vs Task Loss Pointmass   (c) Meta vs Task Loss Reacher

Figure 3: Results of ML$^3$ for MBRL. We can see that, for MBRL the meta-loss generalizes well (a) and speeds up learning when compared to the task-specific loss during meta test. The results are shown for the PointmassGoal (a and b) and the ReacherGoal (c) environments.

call the ReacherGoal environment (see Appendix A.2 for details). The task distribution $p(\mathcal{T})$ consists of different target positions that either the point mass or the arm should reach. During meta-train time, a model of the system dynamics, represented by a neural network, is learned from samples of the currently optimal policy. The task loss during meta-train time is $\mathcal{L}_{\mathcal{T}}(\theta) = \mathbb{E}_{\pi_\theta, P}[R(\tau)]$, where $R(\tau)$ is the final distance from the goal $g$, when rolling out $\pi_{\theta_{\text{new}}}$ in the dynamics model $P$. Taking the gradient $\nabla_\phi \mathbb{E}_{\pi_{\theta_{\text{new}}}, P}[R(\tau)]$ requires the differentiation through the learned model $P$. The input to the meta-network is the state-action trajectory of the current roll-out and the desired target position. The meta-network outputs a loss signal together with the learning rate to optimize the policy. Fig. 3a shows the qualitative reaching performance of a policy optimized with the meta loss during test on PointmassGoal. The meta-loss network was trained only on tasks in the right quadrant and tested on the tasks in the left quadrant of the $x, y$ plane, showing the generalization capability of the meta loss. Figure 3b and 3c show a comparison in terms of final distance to the target position at test time. The performance of policies trained with the meta-loss is compared to policies trained with just the task loss, in this case final distance to the target. The curves show results for 10 different goal positions (including goal positions where the meta-loss needs to generalize). When using the task loss alone, we use the dynamics model learned during the meta-train time, as in this case the differentiation through the model is required during test time. As mentioned in Section 3.2.1, this is not needed when using the meta-loss.

### 4.1.3 LEARNING REWARD FUNCTIONS FOR MODEL-FREE REINFORCEMENT LEARNING

In the following, we move to evaluating on model-free RL tasks. Fig. 4 shows results when using two continuous control tasks based on OpenAI Gym MuJoCo environments (Gym, 2019): ReacherGoal and AntGoal (see Appendix A.3 for details).

(a) ReacherGoal              (b) AntGoal              (c) ReacherGoal              (d) AntGoal

Figure 4: (a+b) Policy learned with ML$^3$ loss compared to PPO performance during *meta-test* time. (c+d) Using the same ML$^3$ loss, we can optimize policies of different architectures, showing that our learned loss maintains generality. Each curve is an average over ten different tasks.

Fig. 4a and Fig. 4b show the results of the meta-test time performance for the ReacherGoal and the AntGoal environments respectively. We can see that ML$^3$ loss significantly improves optimization speed in both scenarios compared to PPO. In our experiments, we observed that on average ML$^3$ requires 5 times fewer samples to reach 80% of task performance in terms of our metrics for the model-free tasks.

To test the capability of the meta-loss to generalize across different architectures, we first meta-train our meta-loss on an architecture with two layers and meta-test the same meta-loss on architectures with varied number of layers. Fig. 4 (c+d) show meta-test time comparison for the ReacherGoal and the AntGoal environments in a model-free setting for four different model architectures. Each curve shows the average and the standard deviation over ten different tasks in each environment. Our comparison clearly indicates that the meta-loss can be effectively re-used across multiple architectures

with a mild variation in performance compare to the overall variance of the corresponding task optimization.

### 4.2 SHAPING LOSS LANDSCAPES BY ADDING EXTRA INFORMATION AT META-TRAIN TIME

This set of experiments shows that our meta-learner is able to learn convex loss functions for tasks with inherently non-convex or difficult to optimize loss landscapes. Effectively, the meta-loss allows eliminating local minima for gradient-based optimization and creates well-conditioned loss landscapes.

#### 4.2.1 SHAPING LOSS FOR REGRESSION

We start by illustrating the loss shaping on an example of sine frequency regression where we fit a single parameter for the purpose of visualization simplicity.

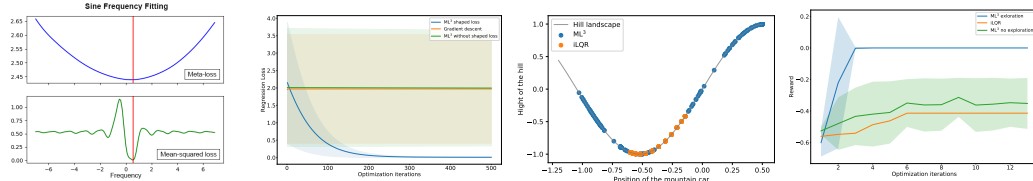

Figure 5: *Left:* Comparison of learned meta-loss (top) and mean-squared loss (bottom) landscapes for fitting the frequency of a sine function. The red lines indicate the target values of the frequency, comparison of optimization performance during meta test time, with and without shaped loss compared to regular gradient descent. *Right:* Improved exploration behavior in the MountainCar environment when using ML[3] with intermediate goals during meta-train time and average distance to the goal at the final timestep.

Fig. 5 (left) shows loss landscapes for fitting the frequency parameter $\omega$ of the sine function $f(x) = \sin(\omega x)$. Below, we show the landscape of optimization with mean-squared loss on the outputs of the sine function using 1000 samples from the target function. The target frequency $\nu$ is indicated by a vertical red line, and the mean-squared loss is computed as $\frac{1}{N} \sum_{i=0}^{N} (sin(\omega x_i) - sin(\nu x_i))^2$. As noted by Parascandolo et al. (2017), the landscape of this loss is highly non-convex and difficult to optimize with conventional gradient descent. In our work, we can circumvent this problem by introducing additional information about the ground truth value of the frequency at meta-train time, however only using samples from the sine function at inputs to the meta-network. That is, during the meta-train time, our task-specific loss is the squared distance to the ground truth frequency: $(\omega - \nu)^2$. The inputs of the meta-network are the target values of the sine function: $sin(\nu x_i)$, similar to the information available in the mean-squared loss. Effectively, during the meta-test time we can use the same samples as in the mean-squared loss, however achieve convex loss landscapes as depicted in Fig. 5 (left) at the top. When comparing the performance of the shaped loss with a regular loss, it becomes evident that without shaping the loss landscape, the optimization is prone to getting stuck in a local optimum, similar to the optimization with regular gradient descent.

By providing additional reward information during meta-train time, as pointed out in Section 3.3, it is possible to shape the learned reward signal such that it improves the optimization during policy training. By having access to additional information during meta-training, the meta-network can learn a loss function that provides exploratory strategies to the agent during test time. In Fig. 5 (right) we compare the performance of a policy trained with the learned loss with and without extra information during train time, showing that adding intermediate rewards during train time shapes the loss landscape such that it facilitates optimization during test.

#### 4.2.2 SHAPING LOSS FOR RL

We analyze loss landscape shaping on the MountainCar environment (Moore, 1990), a classical control problem where an under-actuated car has to drive up a steep hill. The propulsion force generated by the car does not allow steady climbing of the hill, thus greedy minimization of the distance to the goal often results in a failure to solve the task. The state space is two-dimensional consisting of the position and velocity of the car, the action space consists of a one-dimensional torque. In our experiments, we provide intermediate goal positions during meta-train time, which

are not available during the meta-test time. The meta-network incorporates this behavior into its loss leading to an improved exploration during the meta-test time as can be seen in Fig. 5-3, when compared to a classical iLQR-based trajectory optimization (Tassa et al., 2014). Fig. 5-4 shows the average distance between the car and the goal at last rollout time step over several iterations of policy updates with ML$^3$ and iLQR. As we observe, ML$^3$ can successfully bring the car to the goal in a small amount of updates, whereas iLQR is not able to solve this task.

### 4.2.3  EXPERT INFORMATION DURING TRAIN TIME

Expert information, like demonstrations for a task, is another way of adding relevant information during meta-train time, and thus shaping the loss landscape. In *learning from demonstration (LfD)* (Pomerleau, 1991; Ng et al., 2000; Billard et al., 2008), expert demonstrations are used for initializing robotic policies. In our experiments, we aim to mimic the availability of an expert at meta-test time by training our meta-network to optimize a behavioral cloning objective at meta-train time. We provide the meta-network with expert state-action trajectories during train time, which could be human demonstrations or in our experiments trajectories optimized using iLQR. During meta-train time, the task loss is the behavioral cloning objective $\mathcal{L}_\mathcal{T}(\theta) = \mathbb{E}\left[\sum_{t=0}^{T-1}[\pi_{\theta_{\text{new}}}(a_t|s_t) - \pi_{\text{expert}}(a_t|s_t)]^2\right]$. Fig. 6 shows the results of our experiments in the ReacherGoal environment.

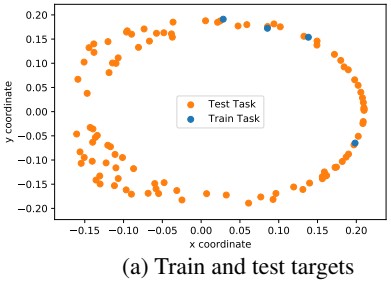
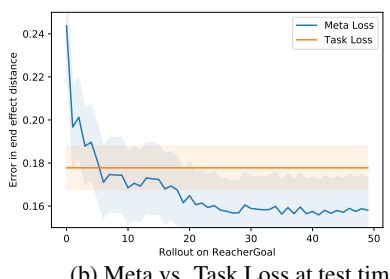

(a) Train and test targets       (b) Meta vs. Task Loss at test time

Figure 6: ReacherGoal with expert demonstrations available during meta-train time. (a) shows the targets in end-effector space. The four blue dots show the training targets for which expert demonstrations are available, the orange dots show the meta-test targets. In (b) we show the reaching performance of a policy trained with ML$^3$ at meta-test time, compared to the performance of training simply on the behavioral cloning objective and testing on test targets.

## 5  CONCLUSIONS

In this work we presented a framework to meta-learn a loss function entirely from data. We showed how the meta-learned loss can become well-conditioned and suitable for an efficient optimization with gradient descent. When using the learned meta-loss we observe significant speed improvements in regression, classification and benchmark reinforcement learning tasks. Furthermore, we showed that by introducing additional guiding information during training time we can train our meta-loss to develop exploratory strategies that can significantly improve performance during the meta-test time.

We believe that the ML$^3$ framework is a powerful tool to incorporate prior experience and transfer learning strategies to new tasks. In future work, we plan to look at combining multiple learned meta-loss functions in order to generalize over different families of tasks. We also plan to further develop the idea of introducing additional curiosity rewards during training time to improve the exploration strategies learned by the meta-loss.

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

## A    APPENDIX

### A.1    MFRL AND MBRL ALGORITHMS DETAILS

---

**Algorithm 3** ML$^3$ for MBRL *(meta-train)*

1: $\phi, \theta \leftarrow$ randomly initialize parameters
2: Randomly initialize dynamics model $P$
3: **while** not done **do**
4:    $\tau \leftarrow$ forward unroll $\pi_\theta$ using $P$
5:    $\pi_{\theta_\text{new}} \leftarrow$ optimize$(\tau, \mathcal{M}_\phi, g, R)$
6:    $\tau_\text{new} \leftarrow$ forward unroll $\pi_{\theta_\text{new}}$ using $P$
7:    *Update $\phi$ to maximize reward under $P$:*
8:    $\phi \leftarrow \phi - \eta\nabla_\phi\mathcal{L}_\mathcal{T}(\tau_\text{new})$
9:    $\tau_{real} \leftarrow$ roll out $\pi_{\theta_\text{new}}$ on real system
10:   $P \leftarrow$ update dynamics model with $\tau_{real}$

---

**Algorithm 4** ML$^3$ for MFRL *(meta-train)*

1: $\phi, \theta \leftarrow$ randomly initialize parameters
2: **while** not done **do**
3:    $\pi_{\theta_\text{new}} \leftarrow$ optimize$(\pi_\theta, \mathcal{M}_\phi, g, R)$
4:    $\tau_{new} \leftarrow$ roll out new policy $\pi_{\theta_\text{new}}$
5:    $\phi \leftarrow \phi - \eta\nabla_\phi\mathcal{L}_\mathcal{T}(\tau_{new})$

---

**Algorithm 5** ML$^3$ for RL *(meta-test)*

1: Randomly initialize policy $\pi_\theta$
2: **for** $j \in \{0, \dots, M\}$ **do**
3:    $\tau \leftarrow$ roll out $\pi_\theta$ in environment
4:    $\theta \leftarrow \theta - \alpha\nabla_\theta\,\mathbb{E}\left[\mathcal{M}_\phi(s, \tau, R)\right]$

---

We notice that in practice, including the policy's distribution parameters directly in the meta-loss inputs, e.g. mean $\mu$ and standard deviation $\sigma$ of a Gaussian policy, works better than including the probability estimate $\pi_\theta(a|s)$, as it provides a direct way to update the distribution parameters using back-propagation through the meta-loss.

## A.2 EXPERIMENTS: MBRL

The forward model of the dynamics is represented in both cases by a neural network, the input to the network is the current state and action, the output is the next state of the environment.

The Pointmass state space is four-dimensional. For PointmassGoal $(x, y, \dot{x}, \dot{y})$ are the 2D positions and velocities, and the actions are accelerations $(\ddot{x}, \ddot{y})$.

The ReacherGoal environment for the MBRL experiments is a lower-dimensional variant of the MFRL environment. It has a four dimensional state, consisting of position and angular velocity of the joints $[\theta_1, \theta_2, \dot{\theta}_1, \dot{\theta}_2]$ the torque is two dimensional $[\tau_1, \tau_2]$

## A.3 EXPERIMENTS: MFRL

The ReacherGoal environment is a 2-link 2D manipulator that has to reach a specified goal location with its end-effector. The task distribution (at meta-train and meta-test time) consists of initial random link configurations and random goal locations within the reach of the manipulator. The performance metric for this environment is the mean trajectory sum of negative distances to the goal, averaged over 10 tasks. As a trajectory reward $R_g(\tau)$ for the task-loss (see Eq. 6) we use $R_g(\tau) = -d + 1/(d + 0.001) - |a_t|$ , where $d$ is the distance of the end-effector to the goal $g$ specified as a 2-d Cartesian position. The environment has eleven dimensions specifying angles of each link, direction from the end-effector to the goal, Cartesian coordinates of the target and Cartesian velocities of the end-effector.

The AntGoal environment requires a four-legged agent to run to a goal location. The task distribution consists of random goals initialized on a circle around the initial position. The performance metric for this environment is the mean trajectory sum of differences between the initial and the current distances to the goal, averaged over 10 tasks. Similar to the previous environment we use $R_g(\tau) = -d + 5/(d + 0.25) - |a_t|$ , where $d$ is the distance from the center of the creature's torso to the goal $g$ specified as a 2D Cartesian position. In contrast to the ReacherGoal this environment has 33 [3] dimensional state space that describes Cartesian position, velocity and orientation of the torso as well as angles and angular velocities of all eight joints. Note that in both environments, the meta-network receives the goal information $g$ as part of the state $s$ in the corresponding environments.

## A.4 EXPERIMENTS: REGRESSION AND CLASSIFICATION DETAILS

For the sine task at meta-train time, we draw 100 data points from function $y = \sin(x - \pi)$, with $x \in [-2.0, 2.0]$. For meta-test time we draw 100 data points from function $y = A\sin(x - \omega)$, with $A \sim [0.2, 5.0]$, $\omega \sim [-\pi, pi]$ and $x \in [-2.0, 2.0]$. We initialize our model $f_\theta$ to a simple feedforward NN with 2 hidden layers and 40 hidden units each, for the binary classification task $f_\theta$ is initialized via the *LeNet* architecture (Lecun et al., 1998). For both regression and classification experiments we use a fixed learning rate $\alpha = \eta = 0.001$ for both inner ($\alpha$) and outer ($\eta$) gradient update steps. We average results across 5 random seeds, where each seed controls the initialization of both initial model and meta-network parameters, as well as the the random choice of meta-train/test task(s), and visualize them in Fig. 2. Task losses are $\mathcal{L}_{\text{Regression}} = (y - f_\theta(x))^2$ and $\mathcal{L}_{\text{BinClass}} = CrossEntropyLoss(y, f_\theta(x))$ for regression and classification meta-learning respectively.

---

[3]In contrast to the original Ant environment we remove external forces from the state.

