# OpenReview forum: "Meta Learning via Learned Loss"
_ICLR.cc/2020/Conference — Reject_

### Official Review · AnonReviewer2 · 2019-10-22
**Official Blind Review #2**

**Rating:** 3

**Review:**

This paper proposes a new meta-learning method (ML3) that meta-learns a loss function that is able to generalize across tasks. Building upon bi-level optimization framework as in MAML, instead of using a task-specific loss function in the inner loop, the authors compute adapted parameters of the model using a parametrized loss network and learn the loss network via backpropagation. Experiments are conducted on supervised sinusoid regression and binary digit classification as well as on model-based and model-free RL benchmarks.

Overall, this paper is an extension to the gradient-based meta-learning algorithms such as MAML. While the idea is natural, there is a prior work [1] that has investigated the effectiveness of learned loss in gradient-based meta-learning, which seems pretty similar to this paper. I wonder how this method could be compared to [1] in various domains.

Besides, I wonder how important the extra information added during the meta-training time is and the authors should present comparison to ML3 without the extra information.

Moreover, I believe comparing ML3 to more recent meta-learning algorithms such as various MAML variants (e.g. MAML++), PEARL, LEO, etc. would be important to show the effectiveness of ML3. Right now, the method is only compared to ML3 with task loss, which seems not very conclusive.

[1] Yu, T., Finn, C., Xie, A., Dasari, S., Zhang, T., Abbeel, P., & Levine, S. (2018). One-shot imitation from observing humans via domain-adaptive meta-learning. arXiv preprint arXiv:1802.01557.

**Experience Assessment:**

I have published in this field for several years.

**Review Assessment: Checking Correctness Of Derivations And Theory:**

N/A

**Review Assessment: Checking Correctness Of Experiments:**

I assessed the sensibility of the experiments.

**Review Assessment: Thoroughness In Paper Reading:**

I read the paper at least twice and used my best judgement in assessing the paper.

---

> ### Author Response · Authors · 2019-11-09
> **Re: Official Blind Review #2**
>
> We respectfully disagree with the reviewer about our work being an extension to MAML. We tried to make this clearer in the related work section and would like to ask the reviewer to review our updated draft. The reviewer correctly points out that we present a bi-level optimization framework, as MAML does. However MAML presents a bi-level optimization framework to optimize model parameters given a known loss function. Our work is concerned with learning the parameters of a loss function that is then used to learn model parameters. Note, that most MAML-variants also optimize model parameters and not loss function parameters. Related work such as [1] takes MAML a step further, and learns an additional “adaptation loss” that is trained to adapt MAML-pretrained parameters more effectively at meta-test time. Note that in this work this adaptation loss is constrained to the architectures of their MAML pre-trained models and absolutely requires the expert demonstrations to be trained.
>
> We on the other hand present a general-purpose framework to learn loss landscape of regression/classification/RL settings, such that new models/policies, of varying architectures, can be learned from scratch without expert demonstrations (but can utilize them when available).
>
> We argue that learning loss function parameters is more general than meta-learning model parameter initializations (as MAML does) and have shown evidence for that in our paper:
>
> 1) in the supervised learning experiments we learn the loss function from data drawn from one task, and show generalization performance to a number of unseen tasks
> 2) Our learned loss functions are not tied to the model architectures, thus we can use our loss functions to learn models of different model-architectures (never seen during meta-train), as we show in our RL experiment section
>
> Thus we argue that our approach is rather orthogonal to MAML.
>
> Further a concern of the reviewer is the lack of comparison to the existing meta-learning algorithms, such as MAML and PEARL. We do not believe that such a comparison would be very meaningful due to the very different nature of MAML/PEARL vs our meta-learning framework. First, in our work, we are targeting to test the hypothesis about capability to learn non-handcrafted losses facilitating optimization. We learn the loss purely from data and train a separate neural network for this. Learning a loss is in contrast to learning a good initialization of the parameters (MAML) or a good embedding space (PEARL) as we are learning a loss function that can succeed from any initialization of an optimizee, which is significantly different from learning a single set of parameters or a suitable representation since loss functions can be used across optimizations for different tasks and policy architectures (which we show in our experiments).
>
> More concretely, comparisons between MAML style approaches and ML3 would have to be rather artificial for the following reasons:
>
> 1) ML3 does not depend on having multiple tasks splits at meta-train time, in fact in the supervised learning experiments we learn the loss function from data drawn from one task, and show generalization performance to a number of unseen tasks (showing the power of learning a loss function instead of model parameter initialization).
>
> 2) MAML-style approaches work on the premise of wanting to adapt pre-trained model parameters to a “new” task in very few (K-shot) steps at meta-test time. We, on the other hand, are not concerned with adapting pre-trained model parameters at meta-test time, we are studying whether our learned loss functions can optimize a model/policy from scratch at meta-test time, and can do so more effectively than a hand-designed loss.
>
> 3) Finally, we wanted to test the hypothesis that loss learning can retain knowledge of extra information available during meta-train time, like exploration incentives or demonstrations. This information might not be available during test time because it is expensive to acquire. The results show that the learned loss function, incorporates the extra information and makes it available to the optimizee, during test time, in the form of a shaped loss landscape.
>
> We would like to ask the reviewer if, in their opinion, it would be beneficial to other readers to move a comparison to MAML to the appendix, or even add a section to the main paper making this distinction very clear to avoid confusion in the future.
>
> We also updated Figure 5 to show the benefit of adding additional information during meta train time and would ask the reviewer to look at out updated results.

---

### Official Review · AnonReviewer1 · 2019-10-23
**Official Blind Review #1**

**Rating:** 3

**Review:**

This paper proposes a mete-learning approach to learn a loss function from old tasks which can generalize well to new tasks. The benefits of the proposed approaches are 1) data-driven a loss function and 2) allowing the usage of extra side-information to design the loss function.

Overall, the presentation in this paper is hard for me to understand technical details and see the difference with existing methods. Please see the questions below. I am glad to discuss problems with the authors' reply during the rebuttal.

Q1. "Mφ(y; fθ(x)) that predicts the loss gave the ground truth target y and the predicted target fθ(x)", "the purpose...loss function, or a meta-loss, which generalizes across multiple training contexts or tasks".
- It is better for the authors to visualize all y v.s. f_{\theta}(x) for all experiments. They have done this only for "Section 4.2.1 SHAPING LOSS FOR REGRESSION". It is better to do this for all the experiments. In this way, we can see why meta-loss can better.
- Besides, it is also better to show some example samples where the learning loss is significantly different from the human-designed one. This helps the reader better understand why the meta-loss can better.
- Finally, the authors claim the extra-information used in the meta-training is helpful. How can we see this point? There is not a step-by-step ablation study on this point.

Q2. Except for problem setup, i.e., learning a loss function, what are novelties in using meta-learning techniques?

Q3. The authors present three usages of the proposed framework in Section 3. Could the authors describe one in detail and then briefly mention the other two usages instead of writing them with the same importance? In this way, readers can understand materials and novelties better.
- For example, I do not understand how exactly gradients are updated on meta-level. The description in Section 3.1. is too brief.

Q4. Why the convergence speed of the meta-learner is important? e.g., Figure 4(b-c).

Q5. We have some basic restrictions for "loss function", i.e., loss(x, y) >= 0 for any x, y; loss(x, x) = 0. How such basic requirements are ensured by the learned meta-loss?

Q6. Could the authors add more explanation in the experiments and motivation in the main text? Currently, the authors just describe what they have done in the proposed method and what they have observed in experiments, just a list of facts (see Q1).

**Experience Assessment:**

I have read many papers in this area.

**Review Assessment: Checking Correctness Of Derivations And Theory:**

I did not assess the derivations or theory.

**Review Assessment: Checking Correctness Of Experiments:**

I assessed the sensibility of the experiments.

**Review Assessment: Thoroughness In Paper Reading:**

I read the paper at least twice and used my best judgement in assessing the paper.

---

> ### Author Response · Authors · 2019-11-09
> **Re: Official Blind Review #1**
>
> We would like to thank reviewer 1 for the important feedback and insightful suggestions. We are glad that our efforts on learning a loss function purely from data in a fully differentiable fashion has been appreciated. In the following we would like to answer the reviewers questions.
>
> Q1:
>
> - We agree that visualizing the loss landscape is a powerful analysis tool, however unfortunately, as the number of parameters in the policy/model increases, the visualization becomes very difficult since we have to plot the loss landscape as a function of the policy/model parameters. Our goal with (Fig. 5, previously Fig.6) was to show the reader how different the loss function can look, even for low-dimensional settings..
>
> - You are right, so far the usefulness of extra information is shown only implicitly in Figure 5 - eg optimizing the meta-learned loss function should lead to a solution faster/more robustly. We have added a comparison of learning the sine function parameters with meta-losses that were learned with/without the extra information. We also updated the Mountain Car RL experiment to include a comparison to training a policy with the meta loss that was not trained with extra information. We can see that the learned-loss that was given extra information at meta-train time, optimizes policies more effectively at meta-test time, even though the extra information is not provided at meta-test time. We hope this comparison clarifies the usefulness of extra information during meta train time.
>
> -Generally speaking, adding extra information during meta-train time is helpful when the extra information is expensive to get, for example intermediate goals or demonstrations for a robotic task. Being able to learn a loss function, that incorporates these demonstrations and can be used for other tasks, where it is not necessary to collect these expensive demonstrations, is for example a benefit of adding additional information during meta train time.
>
> Q2:  we believe that meta learning a loss function is a novel and important research direction to explore within our community. Our main contribution is the gradient-based framework for learning general-purpose loss functions. To summarize advantages (contributions) of our framework:
> i) learned loss allows leveraging of extra information during meta-train time that helps shape the loss function to learn models/policy more effectively at meta-test time.
> ii) it allows faster (more sample efficient) learning of new tasks during meta-test time (in both supervised and RL settings) (as compared to using hand-designed losses)
> iii) it allows change of the architecture without the need of retraining meta-optimized part (i.e. meta-loss) that can not be achieved with other approaches, such as MAML and RL^2.
>
> In order to make the contribution clearer we updated the related work section and we would like to ask the reviewer to look at our updated paper draft.
>
> Q3: We apologize if the explanation of our method was unclear to the reviewer and for the confusion this might have caused. Our aim in section 3 was to describe the different possibilities for implementation of our ML3 framework but given the limited space (max 8 pages) we had to restrict ourselves. In our updated draft we tried to make the explanation more crisp and clear, we would like to ask the reviewer to have a look at the updated section 3 and hope that now it is clearer how the parameters are updated at meta level.
>
> Q4: Improving the convergence speed of training a policy in a reinforcement learning task (model free and model based ) is a desirable effect from different perspectives. First, it reduces the sample complexity, e.g the real rollouts in the environment which makes it cheaper to train from a computational perspective. It is especially beneficial from the hardware perspective, i.e. when the experiments are done on a real robot. Second, it allows achieving the desirable behaviour faster in terms of wall time, due to fewer rollouts needed.
>
> Q5: On one hand, we understand the desire of the reviewer to introduce the aforementioned constraints (properties). On the other hand, we respectfully disagree that the mentioned properties are mandatory for the loss functions in general. For that reason we don’t explicitly enforce them. We would like to point out that some of these properties can be captured by an appropriate choice of the loss model. For example, output >= 0 could be captured by a proper choice of an output activation function.
>
> Q6: Due to the limited space we have decided to move the experimental details into the appendix. We will however make every effort during the rebuttal to include more details and explanations in the text after the reviewers remark on clarity. We would like to ask the reviewer to take a look at our updated paper draft.

---

### Official Review · AnonReviewer3 · 2019-10-24
**Official Blind Review #3**

**Rating:** 6

**Review:**

This paper present a meta-learning method for learning parametric loss functions that can generalise across different tasks and model architectures, which is done by encode learning strategies into an adaptive high-dimensional loss.

I think one interesting result is the utilisation of extra information that helps shape the loss landscapes at meta-train time, where as the authors said the extra information can take on various forms, such as exploratory signals or expert demonstrations for RL tasks. This enables a more efficient ways to optimise the original task loss.

Potential improvements:

(1) paper layout, to be honest, I'm not sure if Figure 1 is really needed

(2) related work section seems very long, would be good if it can be shorten and use the extra space for results display


**Experience Assessment:**

I have published one or two papers in this area.

**Review Assessment: Checking Correctness Of Derivations And Theory:**

I assessed the sensibility of the derivations and theory.

**Review Assessment: Checking Correctness Of Experiments:**

I assessed the sensibility of the experiments.

**Review Assessment: Thoroughness In Paper Reading:**

I read the paper at least twice and used my best judgement in assessing the paper.

---

> ### Author Response · Authors · 2019-11-09
> **Re: Official Blind Review #3**
>
> We would like to thank the reviewer for the suggestions and are happy that the reviewer found the work interesting and possibly impactful for future work in this field. We will do our best to address the remarks about the layout and the related work section during this rebuttal to increase the space for result display.
>
> We would be grateful for any further comments/criticism that could help us improve the paper during the rebuttal phase.

---

### Public Comment · ~Wuyang_Chen1 · 2019-10-25
**Maybe a minor typo?**

I wonder if the last word in the fourth line below equation 1 is "outer" instead of "inner"?

Thank you!

---

> ### Author Response · Authors · 2019-10-27
> **Re: Maybe a minor typo?**
>
> Hi, in the case of one inner optimisation step the outer and the inner loop are the same. You could replace 'inner' with 'outer' in this case.  In general: the parameters of the meta-network are updated after each outer iterations, but the gradients of the task loss can be computed after each inner iteration and saved until the update. I hope this  helps!

---

### Author Response · Authors · 2019-11-12
**Additional Feedback/Comments**

Dear reviewers,  we would be very glad for and really appreciate any additional feedback/comments from the reviewers after our rebuttal to further improve the paper during the discussion phase.

---

### Decision · Program_Chairs · 2019-12-19

**Decision:**

Reject

**Comment:**

Despite the new ideas in this paper, reviewers feel that it needs to be revised for clarification, and that experimental results are not convincing.  I have down-weighted the criticisms of Reviewer 2 because I agree with the authors' rebuttal.  However, there is still not enough support among the remaining reviews to justify acceptance.